# The Ovarian Cancer Tumor Immune Microenvironment (TIME) as Target for Therapy: A Focus on Innate Immunity Cells as Therapeutic Effectors

**DOI:** 10.3390/ijms21093125

**Published:** 2020-04-28

**Authors:** Denisa Baci, Annalisa Bosi, Matteo Gallazzi, Manuela Rizzi, Douglas M. Noonan, Alessandro Poggi, Antonino Bruno, Lorenzo Mortara

**Affiliations:** 1Immunology and General Pathology Laboratory, Department of Biotechnology and Life Sciences, University of Insubria, 21100 Varese, Italy; denisa.baci@uninsubria.it (D.B.); m.gallazzi7@studenti.uninsubria.it (M.G.); rizzi.manuela79@gmail.com (M.R.); douglas.noonan@uninsubria.it (D.M.N.); 2Laboratory of Pharmacology, Department of Medicine and Surgery, University of Insubria, 21100 Varese, Italy; a.bosi@uninsubria.it; 3IRCCS MultiMedica, 20138 Milan, Italy; antonino.bruno@multimedica.it; 4UOSD Molecular Oncology and Angiogenesis Unit, IRCCS Ospedale Policlinico San Martino, 16132 Genoa, Italy; alessandro.poggi@hsanmartino.it

**Keywords:** ovarian cancer, innate immune cells, tumor microenvironment, macrophages, innate immune cell targeted therapy

## Abstract

Ovarian cancer (OvCA) accounts for one of the leading causes of death from gynecologic malignancy. Despite progress in therapy improvements in OvCA, most patients develop a recurrence after first-line treatments, dependent on the tumor and non-tumor complexity/heterogeneity of the neoplasm and its surrounding tumor microenvironment (TME). The TME has gained greater attention in the design of specific therapies within the new era of immunotherapy. It is now clear that the immune contexture in OvCA, here referred as tumor immune microenvironment (TIME), acts as a crucial orchestrator of OvCA progression, thus representing a necessary target for combined therapies. Currently, several advancements of antitumor immune responses in OvCA are based on the characterization of tumor-infiltrating lymphocytes, which have been shown to correlate with a significantly improved clinical outcome. Here, we reviewed the literature on selected TIME components of OvCA, such as macrophages, neutrophils, γδ T lymphocytes, and natural killer (NK) cells; these cells can have a role in either supporting or limiting OvCA, depending on the TIME stimuli. We also reviewed and discussed the major (immune)-therapeutic approaches currently employed to target and/or potentiate macrophages, neutrophils, γδ T lymphocytes, and NK cells in the OvCA context.

## 1. Overview on Ovarian cancer

Ovarian cancer (OvCA) is one of the most common gynecologic malignancies [1], and it is characterized by relatively high incidence, poor prognosis, and a very high mortality rate [2]. A large number of patients can be successfully treated by conventional therapeutic strategies before the cancer spreads beyond the ovaries in patients diagnosed at International Federation of Gynecology and Obstetrics (FIGO) stage I. The survival rate significantly decreases after OvCA has metastasized to pelvic organs (stage II), across the pelvic cavity to abdominal organs (stage III), or beyond the peritoneal cavity to distant parenchymal organs (stage IV) [3]. The poor survival rate in OvCA is associated with diagnosis at late stage due to delayed onset of symptoms and lack of proper screening [1]. Indeed, surgery is effective in most cases of early stage (FIGO stages I–IIA) with a 5-year survival rate of around 90%, but more than 70% of patients are diagnosed with advanced disease (FIGO stages III–IV) presenting malignant ascites which is an indicator of poor prognosis.

Approximately 90% of all OvCA cases are of epithelial cell origin and, according to their nature could be classified in distinct subtypes: high- and low-grade serous, endometrioid, clear cell, mucinous carcinomas, malignant Brenner tumors, and mixed histology [4]. High-grade serous OvCA (HGSOC), often diagnosed in stages III (51%) and IV (29%) when the spread to the peritoneum has already occurred, exhibits the highest frequency and aggressiveness [5]. HGSOC has been associated with frequent somatic genetic mutations of the tumor suppressor protein p53 (TP53) [6], accounting for over 95% of cases. Notably, p53 mutations have been correlated with enhanced proinflammatory chemokine levels and inflammatory tumor microenvironment (TME) [7]. Germline mutations are involved in more than one-fifth of OvCA cases, and about 65–85% of hereditary ovarian tumors are related to highly penetrant DNA repair-associated genes like BRCA1 and BRCA2 [8]. Other tumor suppressor genes and oncogenes, including the mismatch repair (MMR) genes in Lynch syndrome and other DNA repair genes (i.e., BARD1, CHEK2, RAD51C, RAD51D, PALB2, and BRIP1) are also known to be involved in the mechanism of hereditary ovarian tumorigenesis [9].

Standard treatments for OvCA-diagnosed patients include surgery and chemotherapy (co-treatment with carboplatin and paclitaxel). Currently targeted therapies under investigation include antiangiogenic agents, poly (adenosine diphosphate-ribose) polymerase (PARP) inhibitors, hormone receptor modulators, and immune checkpoint inhibitors [10]. It has been reported that combination therapy with antiangiogenic antibody bevacizumab and standard chemotherapy does not give a substantial difference in the overall survival compared to chemotherapy alone [11]. While the exploitation of neoadjuvant chemotherapy is an even more expanding option, treatment of HGSOC remains a clinical challenge [12].

Recurrence of remission post-surgery and/or chemotherapy is a major feature of OvCA, as a consequence of the induction of multidrug resistance. Genetic and epigenetic mutations leading to extrusion or inactivation of cytotoxic drugs, impaired apoptosis, and enhanced induction of repair mechanisms are major orchestrators of this process, all together contributing to the poor prognosis of OvCA. Thus, novel therapeutic strategies and biomarkers are urgently needed.

## 2. OvCA Tumor Immune Microenvironment (TIME)

Besides malignant transformed cells, tumors are composed of normal cells including epithelial cells, fibroblasts, muscle cells, and inflammatory immune cells, altogether generating the TIME [13,14,15]. Within this environment and upon tumor-driven stimuli, cancer can generate a tumor-permissive soil by reprogramming cells of the hosts that acquire tumor-supporting phenotypes and functions [13,14,15].

Evidence has demonstrated that OvCA possesses specific metastatic tropism to the omentum, characterized by highly vascularized immune cell structures called milky spots, playing a pivotal role in the creation of the metastatic TME in the intraperitoneal cavity. OvCA peritoneal metastasis is distinctive because of the extraordinary inflammatory and immunosuppressive milieu of the intraperitoneal cavity, associated with accumulation of malignant ascites [16,17]. Research to date has not yet determined the cellular dynamics that establish the premetastatic niche in OvCA and the omentum.

However, it is now recognized that the metastatic step to the omentum and peritoneum is sustained by a TME enriched with pro-tumor soluble factors, migrated cancer cells, anergic and pro-tumor inflammatory/immune cells, and other host cells, supporting tumor cell proliferation, progression, chemoresistance, and immune evasion [18].

Apart from genetic and epigenetic alterations, OvCA is characterized by a unique peritoneal TME that instructs the complex network between tumor cells and immune cells of the hosts [18,19].

After the “genomic era”, a relevant shift to the host cells as target, first with endothelial cells and angiogenesis and recently with impaired immune response as a principal hallmark of many tumors, occurred. This knowledge has launched numerous clinical trials testing immunotherapy, starting with melanoma, lung, colon cancer and also OvCA [20].

Several cells of both innate and adaptive immunity, including tumor-associated macrophages (TAMs), tumor-associated neutrophils (TANs), myeloid-derived suppressor cells (MDSCs), γδ T cells, and natural killer (NK) cells, directly or indirectly (via soluble interactions) shape the peritoneal TME, creating a permissive environment for tumor development and a favorable metastatic soil [18,19].

One of the main reasons for disease progression and treatment failure is the establishment of a complex immune suppression network that effectively neutralizes antitumor activity of TAMs, TANs, γδ T cells, and NK cells. To make matters worse, TAMs, TANs, and NK cells acquire pro-tumor activities by increasing vascular endothelial growth factor (VEGF) and matrix metalloproteinase (MMP) expression, sustaining tumor vascularization metastasis [21,22,23].

Based on this knowledge, it is clear the central role of innate and adaptive immunity in orchestrating tumor cell fate. Indeed, cancer immunotherapy has emerged as one of the most promising approaches in oncology, assuring the (re)activation of the host defenses, a major strategic role in fighting cancers.

A wide literature is available on immune suppressive effector cells within the TIME, such as MDSCs and T regulatory (Treg) cells [24,25,26], but here we will focus on selected players of innate immunity. In this review, we discuss on macrophages, neutrophils, γδ T cells, and NK cells as relevant TIME-drivers in OvCA progression and as targets for immune therapy.

## 3. Macrophages

Circulating monocytes are recruited to the tumor site by different factors, such as CCL2 (MCP-1), macrophage colony stimulating factor-1 (M-CSF/CSF-1), and metabolites of 5-lipoxygenase [27], and differentiate in response to the microenvironment stimuli into classically activated M1 or alternative activated M2 phenotypes. The M1 macrophages are induced by interferon γ (IFNγ) and lipo poly saccharide (LPS) and exhibit antitumor activity, whereas Th2 signals, such as IL-4 and IL-13, polarize monocytes to the M2 subtype that promote tumor development secreting a variety of growth factors as well as angiogenic and immunosuppressive cytokines [28]. The recruitment of macrophages represents one of the crucial events for the ovarian cancer progression and metastasis [29]. Given the heterogeneity of TAM subsets, the role of these cells in OvCA is still widely discussed. Different studies have shown that a high density of M2-polarized macrophages, particularly the CD163^+^ subset, is associated with poor prognosis and worse overall survival in epithelial ovarian cancer (EOC) whereas high M1/M2 macrophages ratio predicted better prognosis [30,31,32]. A recent cohort study conducted on 140 patients with both different histotypes of primary OvCA and ovarian metastasis from other sites demonstrated that primary HGSOC correlates with a prevalence of M1-like TAMs, a higher M1/M2 macrophages ratio, and longer overall survival (OS) than other tumor subtypes [33]. On the other hand, low-grade serous OvCA exhibited a lower density of CD68^+^ M1-like macrophages and an M2-skewed (CD163^+^) phenotype [34].

High levels of CD163^+^ macrophages are also associated with high levels of IL-6 and IL-10 and short survival in HGSOC patients [35]. These data indicate that the differential macrophage polarization may influence the development of different OvCA subtypes, suggesting the relevance of TAM as a prognostic and predictive factor for OvCA. Although the factors responsible for TAM polarization is not completely understood, it is widely accepted that TAM differentiation involved complex crosstalk between tumor cells and macrophages. Co-culture experiments of OvCA cell lines and macrophages revealed much about the interactions between these two cell types.

Within the TME, M2-like TAMs promote cancer cell proliferation through the activation of the MMP9/HB-EGF pathway and epidermal growth factor (EGF) production, leading to the stimulation of VEGF signaling that supports tumor metastasis [36] (Figure 1).

In addition, co-culture of stage III human ovarian adenocarcinoma cells (IGROV1) and polarized M2 macrophages resulted in enhanced invasiveness of tumor cells, increasing the activity of NF-kB-p65 and c-Jun [37]. Since the blockade of two NF-kB and JNK II-dependent factors, such as extracellular matrix metalloproteinase inducer (EMMPRIN) and macrophage migration inhibitory factor (MIF), reduced the invasiveness of the tumor cell lines, the authors suggested their involvement in this context. Particularly, IGROV1 upregulation of EMMPRIN appeared to induce MMP9 and MMP2 expression in macrophages. Similarly, MIF promotes tumor cell invasiveness, inducing macrophage secretion of MMP [37]. In line with these results, co-culture of the monocytic cell line (THP-1) and BRCA2-mutated ovarian adenocarcinoma cell line (PEO-1) showed that the tumor-derived matrix metalloproteinase-stimulating factor (MMPSF) among with autocrine monocyte-derived TNFα stimulate the production of pro-MMP-9 [38]. In addition, TAMs can induce OvCA invasion, upregulating scavenger receptor SR-A [39]. Indeed, macrophages from SR-A^-/-^ mice co-cultured with the mouse ovarian epithelial papillary serous adenocarcinoma cells (ID8) showed a reduced ability to promote tumor cell invasion. The inhibition of SR-A by 4F peptide, an apolipoprotein A-1 mimetic, was able to counteract tumor growth in an in vitro assay as well as in a mouse model of OvCA, providing support for the SR-A involvement [40]. Together with the previous antitumor effect of subcutaneous and oral administration of 4F in C57BL/6 mice [40], these data suggest a novel potential therapeutic target.

Based on their role in tumorigenesis, TAM depletion emerged as a promising target for OvCA treatment. Germano and colleagues demonstrated that the DNA binder agent trabectedin, approved for the treatment of platinum-sensitive OvCA, causes a significant and selective reduction of tumor macrophages in both the syngeneic (ID8) and xenograft (IGROV) models of OvCA [41]. In the same animal models, the G5-methotrexate nanoparticles targeting folate receptor-2 (FOLR-2) was proposed as the alternative way to deplete TAM and showed greater effectiveness than cisplatin [42]. Currently, many other different approaches acting on TAM have been developed and some are available for clinical evaluation. One of the strategies accounts for the inhibition of macrophage recruitment at the tumor site. For this reason, some clinical trials involving the CSF-1R and CCR2 pathways are ongoing. The reduction of infiltrating macrophages due to the blockade of CSF-1 and its receptor (CSF-1R) was shown to enhance the anti-tumor effect of docetaxel in a mouse model of epithelial OvCA [43] (Figure 2). In another syngeneic mouse model of late-stage EOC, selective inhibition of CSF-1R significantly reduced ascites accumulation and M2 macrophages infiltration, thus reversing the correlated vascular dysfunction [44]. Likewise, the reduction of CCL2 expression by the natural product 9-hydroxycanthin-6-one has been reported to inhibit macrophage recruitment, as well as to counteract the ovarian conditioned medium-driven M2-like phenotype [45] (Figure 2A). Indeed, limitation of M2 polarization represents a second strategy to target TAM. A gene expression analysis showed that the antitumor agent paclitaxel, used for the treatment of several solid tumors, including OvCA, can reinstruct the M2 signature of TAM toward the M1 phenotype [46]. Since, in TLR4 KO mice, paclitaxel-induced M1 polarization was abolished, this effect seems to involve TLR4 activation [46]. Further, as described above for the scavenger receptor SR-A (CD204), M2 macrophages upregulate the mannose receptor (CD206). Human antibody against CD206 has been demonstrated to prevent in vitro TAM polarization by blocking the crosstalk between the HGSOC cell line OVCAR5 and differentiated CD206^low/high^ macrophages via tumor-released mesothelin [47]. TAM repolarization may be also induced by natural compounds, such as Onionin A, from onions and Deoxyschizandrin, from berries, that can abrogate pro-tumor activation of M2 macrophages and can reduce the production of the tumor-promoting factors (MMP9, CCL5, and VEGF) [48] (Figure 2A).

Another macrophage-oriented treatment approach consists of the retrieval of the existing antitumor response, acting on immune checkpoint molecules such as PD-L1 and B7-H4. The programmed cell death-1 (PD-1) is a cell surface receptor that binds two PD-1 ligands (PD-L1 and PD-L2) and negatively regulates T cell activation, suppressing the immune response [49]. Many tumors, including OvCA, can escape host immunity through the PD-1 pathway [50]. Recently, it has been demonstrated that TAMs express PD-L1 in primary epithelial OvCA and metastatic HGSOC [51]. Whereas, in primary epithelial OvCA, PD-L1 is predominantly expressed on immune cells, in recurrent epithelial OvCA, the expression extends to both immune and cancer cells [52,53]. Interestingly, analysis of TME in ovarian platinum-sensitive recurrent cancer revealed higher “classically activated” M1-like macrophage density and higher CD25^+^ Treg cells than in primary tumor, which correlate with long overall survival. Despite the role of PD-L1 as a negative regulator of antitumor response, the data suggested that the dynamics between TAM PD-L1 expression and cytolytic and regulatory T cell subsets consist in a mechanism of adaptive resistance, resulting in an immunological stalemate. In such condition, long-termed immunotherapy may be required [51].

Similar to PD-L1, B7-H4 is highly expressed on the surface of TAMs in high-grade OvCA [54]. Its overexpression correlates with the presence of IL-6 and IL-10 in TME, and it is inhibited by GM-CSF and IL-4 [55]. B7-H4^+^ macrophages counteract T cell-proliferation, cytokine production, and cytotoxicity, suggesting its involvement in the tumor immune escape mechanism [55]. Furthermore, another crucial role is played by CD47, an integrin highly expressed by OvCA cells, which correlates with poor prognosis in OvCA. CD47, interacting with thrombospondin-1 (TSP-1) and signal regulatory protein alpha (SIRPα), negatively regulates macrophage activity [56]. Anti-CD47 mAb treatment was shown to increase in vitro macrophage phagocytosis, to inhibit tumor formation, and to enhance macrophage infiltration in xenograft model of ovarian clear cell adenocarcinoma (SKOV-3) (Figure 2A). In addition, CD47 was highly expressed on the CD133^+^ tumor-initiating population and was able to trigger their phagocytosis [57]. Therefore, given their role as suppressive regulators of the immune response in OvCA, blocking B7-H4 and CD47 molecules or depleting this suppressor cell subtype of TAM have been proposed as promising targets to prevent metastasis and recurrence of OvCA. Thus, the strategy to exploit TAM by repolarization and deletion by killing or blocking of recruitment in combination with immune checkpoint inhibitors represents an attractive target to switch macrophages response in favor of ovarian cell death.

## 4. Neutrophils

Neutrophils represent the first line of defense of the innate immunity against insults from pathogens [58]. Mobilization of neutrophils from the bone marrow depends on a series of stimulating factors and cytokines, including IL-23, IL-17, G-CSF, and CXC chemokine receptors [59]. The detection of neutrophils within the TME is an indirect parameter of inflammation, which is a hallmark of cancer [60]. However, in the cancer scenario, neutrophils have been considered as inert bystanders due to their very short circulating lifespan [61,62]. Evidence now shows that TME extends the neutrophil lifespan and manipulates their differentiation state, generating diverse phenotypic and functional subsets: antitumor “N1-neutrophils” and pro-tumor “N2-like/tumor-associated neutrophils” (TANs). TAN activities in cancer are complex and context dependent [63]. Like TAMs, TANs can exert antitumor or pro-tumor functions [64,65], depending on the tumor microenvironment and related stimuli [63,64,66]. Gene expression analysis revealed TAN-specific signatures associated with tumor resistance to radiotherapy and predicted poor patient survival in 25 malignancies [67]. CXCR1 and CXCR2 are the major neutrophil receptors that mediate neutrophil chemotaxis to the TME in response to G-CSF, GM-CSF, CXCL1, CXCL2, CXCL5, CXCL8, and MIP-1α [61,68] (Figure 1). Neutrophils promote tumor initiation, growth, proliferation, and metastatic spreading through release of growth factors, myeloperoxidases ROS [61,69], VEGF, and MMP9 [70] (Figure 1). Finally, neutrophils facilitate tumor proliferation by enhancing immunosuppression and by attenuating the immune system response. Increased production of ROS, inducible nitric oxide synthase (iNOS), or arginase 1 (ARG1) released by neutrophils suppress CD8^+^ T lymphocyte antitumor activities [66] and can reduce NK cell function [71], thereby facilitating the extravasation of tumor cells and promoting tumor growth and metastasis (Figure 1).

In OvCA, neutrophils are critical players and have been proposed as new potential biomarkers of disease outcome or as therapeutic targets [72]. Meta-analysis studies on systemic inflammatory markers suggest that neutrophil–lymphocyte ratio (NLR) can potentially serve as a prognostic biomarker in ovarian cancer patients [73,74].

An elevated NLR is a predictive factor both for early stage [75] and for poor overall survival in advanced stage of OvCA [30,73,76]. These findings suggest a role of neutrophil influx into the premetastatic niche during early-stage ovarian tumors. Results from recent studies highlight new insight about neutrophils as a “mobile fertilizer” [77] that establish the premetastatic omental niche via neutrophil extracellular traps (NETs). NETs have been found in the omentum of ovarian tumor-bearing mice before metastasis and of patients with early-stage OvCA [78]. Thus, NETs represent an early response by neutrophils to the presence of OvCA, with a key role in the generation of the “fertile” and permissive niche for the subsequent colonization of ovarian circulating cancer cells [77,78]. Accordingly, blockade of peptidyl-arginine deiminase 4 (PAD4), an essential enzyme for NET generation, decreased omental metastasis in orthotopic OvCA models [78] (Figure 2B). NETosis depends on the generation of ROS, CXCL-8, G-CSF, growth-regulated oncogene (GRO) α, or GROβ factors released by early-stage ovarian tumors that stimulate the influx of neutrophils into the premetastatic omental niche [78] (Figure 1). Neutrophil elastases and MPPs of NETs favor the induction of an immunosuppressive microenvironment and reactivate the tumorigenic program in dormant/cancer stem cells [62] (Figure 1). Mitochondrial damage-associated molecular patterns (DAMPs) also may activate neutrophil and generate NETs, thus facilitating metastasis and obstruct antitumor immunity in OvCA [79] (Figure 1).

Neutrophils can modulate and suppress the activity of T cells in the OvCA microenvironment [80,81] and can facilitate the immune escape of OvCA cells by upregulating the expression of PD-L1 [80] (Figure 1). Another recent study confirmed that NLR ratio positively correlates with increased Th17/IL-17 levels and expression of PD-L1 in ovarian carcinoma [82].

Since neutrophils act as pro-tumor immune cells in many human cancers, strategies intended to reduce their activity have been explored and some have entered clinical evaluation. Therapies targeting CXCLs/CXCR2 axis [83], and G-CSF [84,85], that may reduce the migration and the number of neutrophils in tumors areas, have been evaluated in several preclinical studies and clinical trials. Genetic approaches targeting CXCR2 or employment of pharmaceutical CXCR2 inhibitor SB225002 showed promising results in impeding OvCA peritoneal metastases in SKOV-3 xenograft mouse models [86].

The use of CXCR2 inhibitors reduced both neutrophil recruitment and NETosis, enhancing the survival of mouse models of pancreatic cancer. [87]. Similarly, blockades of IL-23 and IL-17 production which in turn regulates granulopoiesis through G-CSF represent a valid approach to control neutrophil production in vivo [88,89,90].

TANs are driven by TGFβ to acquire a polarized N2-like/TAN phenotype; thus, employing TGFβ receptor blockers could be a strategy to shift from N2 to the acquisition of N1 antitumor phenotype and activities [66] (Figure 2B). Some studies have reported an increased effect of combining CXCR2 inhibitors or anti-Ly6G and checkpoint inhibitors that boost the immune system against tumor growth and dissemination [91] (Figure 2B).

Since resistance to antiangiogenic agents has been linked with neutrophil stimulation, combining antiangiogenic drugs with neutrophil targeting agents may sensitize resistant tumors to therapy with antibodies against VEGF [92]. On the other hand, bevacizumab enhanced clinical outcome in OvCA patients with a high NLR [93] (Figure 2B).

Given the dual role of neutrophils in cancer, the consequences of depleting tumor-promoting and antitumor neutrophils are not fully understood. Enhancing our understanding of the role of neutrophils in OvCA is crucial to identify novel prognostic markers and to develop effective therapeutic strategies.

## 5. γδ T Lymphocytes

The involvement of the immune system in the control of OvCA has been shown, and therapeutic options are based on regulating the ovarian TME to stimulate antitumor effector cells [94,95,96,97]. Among antitumor lymphocytes, besides NK cells and cytotoxic T cells (CTL), γδ T lymphocytes can eliminate OvCA cells [94,95,96,97]. The γδ T lymphocytes can be roughly subdivided into two major subsets: Vδ1 subtype is mainly present in tissues, while Vδ2 is the major subset of γδ T cells in the peripheral blood [89,94,95,97]. Both these two cell subsets share some relevant activating receptors such as NKG2D and DNAM-1 [98] that can recognize counter-ligands on ovarian target cells leading to tumor cell killing. Several inhibitory receptors can play a role in regulating γδ T cell killing such as Killer Ig-like inhibitory receptors and C-type lectin inhibitory receptors [95,97,98,99]. Thus, a strict balance between positive and negative signals delivered during the interaction of γδ T cells and OvCA cells eventually leads to the elimination of target cells.

Epithelial ovarian cancers typically express abundant NKG2DL, which are strongly associated with negative disease outcomes [100,101,102] (Figure 3A).

This unexpected effect can be dependent on the immunosuppressive microenvironment supported by the continuous presence in the extracellular milieu of soluble NKG2DLs, which function as decoy receptors binding the NKG2D molecules on antitumor effector lymphocytes such as γδ T cells, besides NK cells and cytolytic CD8^+^ T cells [103,104]. Alternatively, the expression of NKG2DL can give an advantage to tumor cell growth and expansion [105]. More interestingly, the expression of NKG2D molecules on OvCA cells together with NKG2DL can trigger autonomous stimulation of cancer cells as shown in breast cancer cell lines [105,106]. Thus, NKG2D^+^ ovarian cells can have a stronger ability of self-renewing sphere formation in vitro and tumor generation in vivo in immunodeficient mice than NKG2D negative tumor cells [106].

The γδ T cell-mediated killing can be exploited also using humanized monoclonal antibodies (Hu-mAb) able to elicit the antibody-dependent cellular cytotoxicity (ADCC) (Figure 4A) exerted by immune cells bearing the FcγRI/II/III. Notably, this mechanism is shared by γδ T cells and other effectors of the innate immunity, including NK cells and monocytes/macrophages [95,97,99,107,108]. Thus, γδ T cells can eliminate OvCA cells both through specific activating receptors and by using Hu-mAb if the antigen recognized by these antibodies is expressed on cancer cells (Figure 4A).

Recently, it has been shown that γδ T lymphocytes in several solid tumors, such as colorectal, pancreatic, and breast cancers, can promote tumorigenesis and metastasis [109,110,111,112]. Similarly, it has been reported that γδ T cells present in OvCA specimens can produce IL-17A and can exert an inhibitory effect on cell proliferation of CD4^+^ naïve T cells together with low production of IFNγ and a reduced anti-OvCA cytolysis [113] (Figure 3A). Vδ2 T cells expanded from the peripheral blood (PB) can efficiently eliminate in vitro and in vivo murine models of OvCA cells (Figure 4A). This effect is further increased in the presence of pamidronate, a kind of amino-bisphosphonate (N-BPs) [114] (Figure 4A). This finding indicates that it is possible to trigger anti-OvCA effects using appropriate drugs and Vδ2 T cells. These data are in line with previous observations that liposome-encapsulated N-BPs could trigger the elimination of OvCA cells using PB-expanded Vδ2 T cells (Figure 4A). Importantly, this latter formulation of N-BPs can achieve a higher drug concentration than the free drug; the alteration of the pharmacokinetic properties of N-BPs can favor the delivery at the tumor site, reducing their tropism to bone [115,116].

## 6. Natural Killer cells

NK cells are effector lymphocytes of innate immunity, primarily involved in the recognition and elimination of virus-infected and malignant transforming cells [117,118,119]. NK effector functions include ADCC via CD16; production of perforin and granzymes (resulting in the induction of apoptosis on target cells); and release of antitumor cytokines, such as IFNγ and TNFα [117,118,119]. Two major NK cell subsets have been characterized within PB lymphocytes, according to the expression of the CD56 and CD16 surface antigens. CD56^dim^CD16^+^ NK cells (90–95% of total circulating NK cells) act mainly via perforin/granzymes and ADCC [117,118,119]. CD56^bright^CD16^-^ NK cells (5–10% of total circulating NK cells) act via cytolytic cytokine production [117,118,119]. According to the tissue type and the physiopathological conditions, the frequency and distribution of these two NK cell subsets significantly vary, along with their functional impairment or over-activation [120,121,122,123,124,125].

Among the tissue adaptation of NK cell phenotypes, the decidua represents a very peculiar scenario. Apart from their immunotolerant behavior for the developing fetus, in the decidua (d), NK cells acquire the CD56^bright^CD16^-^ phenotype, characterized by the release of high amounts of pro-angiogenic cytokines (such as VEGF, PlGF, and CXCL8) [86,87]. These decidual NK (dNK) cells account for over 50% of total lymphocytes found in this tissue, and they are necessary for the formation of the spiral arteries needed to assure the proper fetus/maternal interface [86,87].

Alteration of NK cell functions had been observed in preclinical models and patients with several cancers [117,120,121,126,127,128,129,130,131], including OvCA [132,133,134]. On the other hand, several studies supported the use of NK cell therapy in OvCA [132,133,134]. Altered NK cell phenotype and functions in OvCA arise as a consequence of OvCA-associated immunosuppressive cytokines and those produced by the surrounding microenvironment [135].

Macrophage migration inhibitory factor (MIF) overexpression has been found in OvCA (Figure 2B). MIF acts on NK cells by downregulating the expression levels for the NKG2D activation receptor and by increasing expression for the inhibitory checkpoint B7-H6; these features are associated with poor OvCA prognosis [136].

MDSCs [137,138] and Treg cells [139,140] are increased in OvCA TME and release large amounts of IL-10 and TGFβ (Figure 3B). TGFβ is largely produced in the TME, both by tumor, stroma, and immune cells [131,141,142,143]. Apart from immunosuppression, cytolytic NK cells exposed to TGFβ have been demonstrated to acquire the dNK-like, pro-angiogenic CD56^bright^CD16^low^ subsets [143,144] (Figure 3B). Given the similarities between the decidua microenvironment and the TME, it is expected that OvCA, as a consequence of its “embryonic-like” features, will employ altered NK cells for tumor progression and angiogenesis.

Sun et al. demonstrated that NK cells from patients with OvCA can effectively lyse different OvCA cell lines, including the HGSOC cell lines OVCAR3 and CAOV-3, the clear cancer cell line SKOV3, and the ovarian endometrioid carcinoma cell line A2780, and can reduce their migratory activities [134]. In addition, immunotherapy with NK cells resulted in reduced metastasis and prolongs survival in an in vivo model where immune-deficient BALB/c-nude mice were implanted with patient-derived CAOV-3 cells [134]. Moreover, mice treated with NK cells also exhibited increased infiltration of CD4^+^ and CD8^+^ T cells along with larger IFNγ levels [134]. Poznansky et al. showed that expanded CD56^superbright^CD16^+^ NK cells from OvCA patients are cytotoxic against autologous tumor in a patient-derived xenograft murine model [132].

Mucin 16 (MUC16/CA125) has been reported to be highly expressed in OvCA [145,146] and has been recognized as a biomarker to monitor epithelial OvCA and for the differential diagnosis of pelvic masses [147,148]. MUC16 supports tumor cell proliferation and dampens immune cell response. Large amounts of MUC16 have been also detected in the peritoneal fluids (PFs) of patients with OvCA [149,150]. Reduced expression of CD16 has been detected in NK cells from OvCA TME [151]. MUC16 has been found to inhibit CD16 expression in peritoneal fluids (Figure 3B). NK cells derived from healthy donors exhibit the same reduced expression of CD16 when exposed to peritoneal fluids of patients with OvCA that are enriched in MUC16. Interestingly, MUC16 is highly expressed during pregnancy. Similar to OvCA, MUC16 production is elevated in early pregnancy and contributes to the induction of the tolerogenic environment during decidualization acting on dNK cells [152,153]. This reinforces the hypothesis that the immune contexture between the TME and the pregnancy environment share several features both at the cellular and molecular levels, governing the fetus–maternal tolerance and immune evasion by OvCA.

The development of ascites is a major feature of OvCA. Ascites fluids are enriched in cytokines and growth factors that support tumor cell proliferation and anergy in immune cells [154,155]. CD3^-^CD56^+^CD16^+^ NK cells have been found at higher concentration in ascites fluids, as compared to those of peripheral blood of patients with OvCA [151]. Therefore, these NK cells exhibit decreased expression of CD16, along with reduced cell proliferation, cytokine release, and cytolytic activities.

Overproduction of TGFβ has been found in patients with OvCA [76,141,142]. TGFβ is also abundant in ascites fluids from OvCA patients [156]. TGFβ acts as a master switcher of immune suppression in NK cells [157], also by converting cytolytic CD56^dim^CD16^+^ ADCC competent NK cells into CD56^bright^CD16^-/low^IFNγ-producing NK cells [158] (Figure 3B). TGFβ is also implicated in the induction of a pro-angiogenic NK cell phenotype in non-small cell lung cancer and colorectal cancer patients [120,121,126,127,128] (Figure 3B) and thus might contribute to NK cell angiogenic switch in OvCA.

The tumor-related MICA/B and B7-H6 ligands have been frequently found in the peritoneal fluids of patients with OvCA. MICA/B and B7-H6 ligand–NK cell interactions resulted in impaired NK cell functions of NK cells [104,156] due to downregulation of NKp30 and decreased ability to release perforin, granzyme B, and IFNγ [156] (Figure 3B).

Cytokines represent the first line of immunotherapy approaches to boost NK cell antitumor activities. Intraperitoneal administration of IL-2 has been reported to increase NK cell number and activation in vivo. Limitations for use of IL-2 for immunotherapy deal with the fact that IL-2 exhibits very heterogeneous variability of success due to the effect of the tumor burden, prior to therapy administration. Combination approaches represent the best strategies to both combine enhanced activities and to reduce the toxicities related to the use of high doses of selected agents. In this scenario, the use of low doses of IL-2 in combination with 13-cis-retinoic acid has been reported to improve clinical outcome in platinum-sensitive advanced OvCA [159]; indeed, this combination resulted in increased T cell and NK cell frequency [159].

Cancer therapy also impacts priming the immune cell response [160,161]. In this view, Mallmann-Gottschalk and collaborators recently showed that OvCA cells, pretreated with anti- epithelial growth factor receptor (EGFR) Tyrosine Kinase Inhibitors (TKIs), acquired increased drug-sensitivity via NK cell-mediated ADCC. Conversely, reduced cytokine secretion was observed in NK cells following TKI sensitization, suggesting that NK cells are differentially modulated by anti-EGFR TKIs [77].

The use of high doses of IL-2 has been associated with elevated toxicities [162,163,164]. Indeed, immunosuppressive Treg cells exhibit higher affinity with IL-2. To overcome these major issues, the use of immune-cytokines, known as antibody–cytokine fusion proteins, has been employed [162,163,164].

IL-15 has potent effects on NK cell activation [165] (Figure 4B). While less toxic compared to IL-2, to reach clinical effectiveness, higher doses of IL-15 are necessary [166]. Therefore, IL-15 administration overcomes the problem of Treg cell expansion that has a major affinity for IL-2 [167]. Current efforts are focusing on the association of IL-15 with super-agonists that act as cargos for the IL-15–IL-15Ra-subunit in a more stable complex [168]. Diverse clinical trials are currently evaluating the IL-15-primed NK immunotherapeutic properties [169]. In this context, elevated activation of NK cells in OvCA ascites has been reported, following interactions with monomeric IL-15 or the IL-15 super-agonist fusion complex ALT-803 [170,171].

Adoptive transferred NK cells, following pre-activation with IL-2, IL-15, and IL-18, have been reported to display increased antitumor activities in OvCA in vitro and in vivo (Figure 4B), and they can also persist in OvCA ascites fluids. NK cells exposed to the IL-15 super-agonist ALT-803 increase their degranulation activities against OvCA cells in vitro, and ascites from OvCA patients can release higher amounts of IFNγ.

Another source of NK cells for immunotherapy is represented by adult cell population, namely hematopoietic stem cells (HSCs) from bone marrow (BM), PB, or cord blood (CB) [121,172,173,174], following ex vivo expansion, cytokine activation. and patient reinfusion (Figure 4B). Induced pluripotent stem cells (iPSCs) have been also explored as a source for NK cell production [175]. Li et al. recently demonstrated that iPSC-NK cells, endowed with a chimeric antigen receptor (CAR) containing the transmembrane domain of NKG2D, the 2B4 co-stimulatory domain, and the CD3ζ signaling domain, significantly inhibited tumor growth and prolonged survival compared to PB-NK cells in a xenograft model of OvCA. Therefore, NK-CAR-iPSC-NK exhibited similar in vivo activity as T-CAR-expressing T cells, though with less toxicity [175].

Several studies confirmed that CAR-engineered immune cells are powerful tools for cancer immunotherapy, and a plethora of CAR have been produced [176,177,178]. This approach has been characterized by a T-cell centric view. Considering their innate killing activities, it is clear that NK cells may be better CAR drivers, and many studies have been recently oriented toward this direction [179,180]. CD24 is a very promising target, which is rarely expressed in normal human tissue and is predominantly found on hematologic cells, and antibody-based treatments against CD24 have been evaluated in preclinical studies [181,182,183].

Lastly, Klapdor et al. recently showed that the human NK-cell line NK-92 with the anti-CD24 CAR showed high cytotoxic activity against CD24-positive OvCA cell lines (SKOV3 and OVCAR3) (Figure 4B). This effect was restricted to CD24-expressing cells [184].

## 7. Conclusions

The lack of curative treatment and the high relapse rate for women with advanced OvCA still represent an urgent unmet clinical problem, highlighting a clear need for new therapeutic strategies. As for some other cancers, immunotherapy approaches have not been as successful as expected. Indeed, the OvCA TME displays unique features leading to immune suppression and tolerance; this impairs both the presence and the activity of immune system components including TAMs, neutrophils, γδ T cells, and NK cells. The design of novel effective therapies should take into account this immunosuppression and should boost up the antitumor activity of the immune system. It is clear that some TIME components have a dual role in the development and progression of OvCA. Further studies can aid to plan effective strategies for the eradication of cancer cells by targeting innate immunity effectors. A better knowledge of the molecular mechanisms regulating the interaction between cancer cells and different cell subsets of innate immune system within the OvCA TME may provide new targets to relieve the impairment of immune response. This approach can contribute to the discovery of effective immunotherapeutic strategies aimed to stimulate the innate arm of immune system in combination with standard therapies for OvCA.

## Figures and Tables

**Figure 1 ijms-21-03125-f001:**
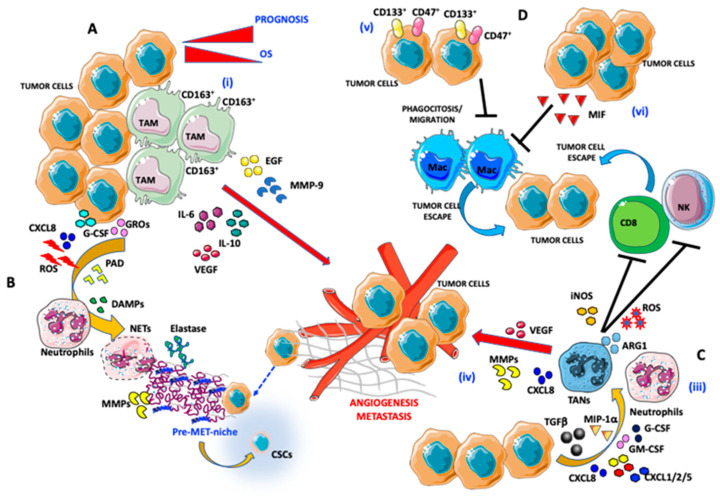
Pro-angiogenic and pro-metastatic tumor immune microenvironment (TIME) landscape in ovarian cancer: focus on macrophages and neutrophils. (**A**) Interaction between tumor-associated macrophages (TAMs) and ovarian cancer (OvCA) cells: (i) CD163^+^ TAMs, by secreting epidermal growth factor (EGF), IL-6, IL-10, vascular endothelial growth factor (VEGF), and matrix metalloproteinase (MMP)-9 trigger angiogenesis and metastatic dissemination. (**B**,**C**) Interaction between tumor-associated neutrophils (TANs) and OvCA cells: (ii) OvCA cells by releasing CXCL8, g-colony stimulating factor (CSF), growth-regulated oncogenes (GROs), and reactive oxygen species (ROS) cooperate with peptidyl arginine deiminase (PAD) and damage-associated molecular patterns (DAMPs) in inducing neutrophil extracellular trap (NET) formation by neutrophils. NET induction is associated with elastase and MMP release that instruct a tumor permissive metastatic niche formation that includes the cancer stem cell (CSC) components and tumor-escape. (iii) Neutrophils are polarized into TANs through transforming growth factor β (TGFβ), macrophage inflammatory protein (MIP)-1a, CXCL1/2/5/8, GM-CSF, and G-CSF secretion by OvCA cells. (iv) Induction of TAN results in promotion of angiogenesis (via VEGF and CXCL8 production) and natural killer (NK) cells and CD8 anergy in an iNOS/ARG1/ROS-dependent manner. (**D**) OvCA/TAM interaction in the TIME supporting tumor cell escape: (v) CD133- and CD47-expressing OvCA cells act on macrophages by blocking their migration in tumor tissues and phagocytosis. (vi) Macrophage migration inhibitory factor (MIF) released by OvCA cells also blocks efficient phagocytosis by macrophages.

**Figure 2 ijms-21-03125-f002:**
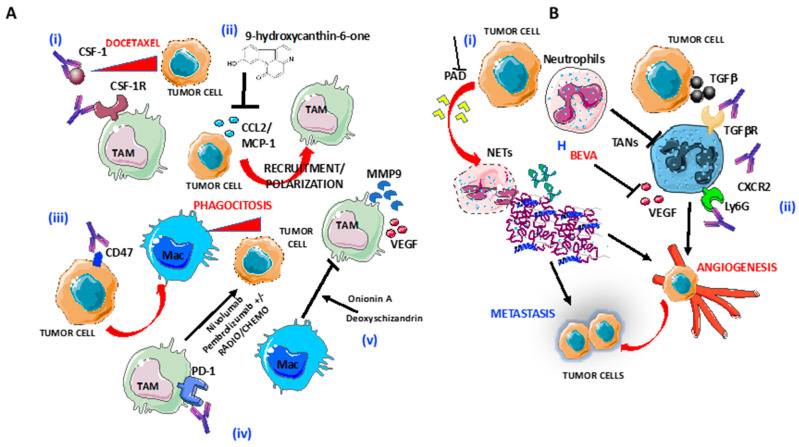
Targeting of macrophages and neutrophils in OvCA: (**A**) Several strategies to target tumor-associated macrophages (TAMs) are shown: (i) blocking of CSF-1R by anti-CSF1 mAbs induce increased responsiveness of tumor cells to Docetaxel; (ii) blocking of CCL2 release by 9-hydroxycanthin-6-one inhibit TAMs recruitment and polarization; (iii) selective blocking of CD47 on tumor cells macrophage infiltration and phagocytosis of ovarian tumor cells; (iv) anti-PD1 Nivolumab and Pembrolizumab in combination with radiotherapy and chemotherapy enhance tumor cells lysis; and (v) Blocking of M2-like/TAM polarization by as Onionin A, from onions and Deoxyschizandrin, from berries. (**B**) Several strategies to target tumor-associated neutrophils (TANs) are shown: (i) blocking of peptidyl-arginine deaminase (PAD) release by tumor cells inhibits neutrophils extracellular trap formation and (ii) preventing TAN polarization by blocking Ly6G and CXCR2 in combination with checkpoint inhibitors, TGFβR and VEGF, through Bevacizumab.

**Figure 3 ijms-21-03125-f003:**
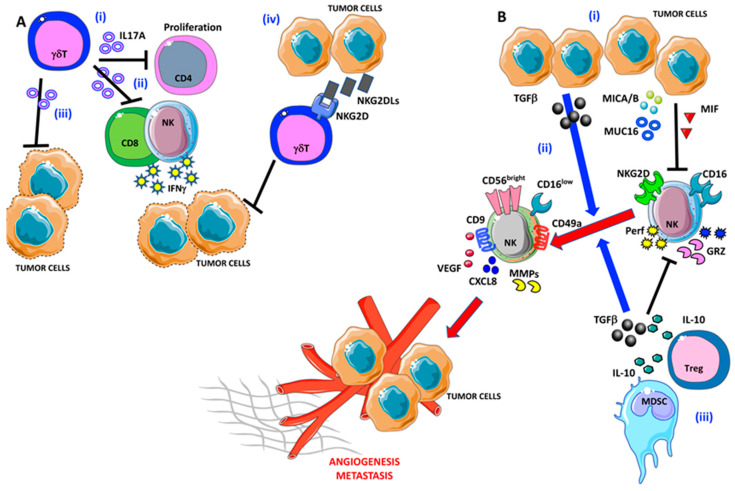
Pro-angiogenic and pro-metastatic TIME landscape in ovarian cancer: focus on γδ T and NK cells. (**A**) (i) γδ T cells are able to block CD4 T cell proliferation and (ii) interferon γ (IFNγ) release by NK cells and CD8^+^ T cells via IL-17A. (iii) IL-17A-producing γδ T cells are not efficient in tumor cell lysis. Moreover, OvCA cells are able to produce large amounts of NKG2DLs that interfere with tumor cell elimination by γδ T cells (iv). (**B**) (i) OvCA released several factors that are involved in NK cell polarization, in particular TGFβ, MICA/B, MUC16, and MIF that induce NK cell anergy by decreasing their ability to produce perforin and granzymes. (ii) Besides, TGFβ participates in the pro-angiogenic switch of NK cells that acquire the decidual-like phenotype and functions. (iii) Finally, myeloid-derived suppressor cells (MDSCs) cooperate with the OvCA cells to NK cell anergy/acquisition of pro-angiogenic feature via TGFβ and IL-10.

**Figure 4 ijms-21-03125-f004:**
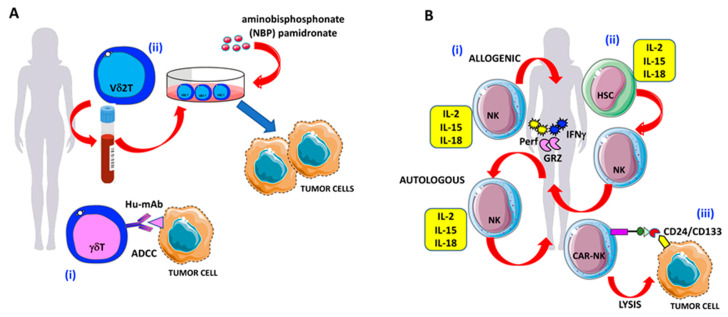
γδ T and NK cell-based therapies against OvCA cells: (**A**) Strategies to enhance γδ T cell antitumor activities in OvCA. (i) Triggering of γδ T cell-mediated antibody-dependent cellular cytotoxicity (ADCC) by specific hu-mAbs. (ii) Expansion of Vδ2 T cells derived from the peripheral blood in the presence of the amino-bisphosphonates (N-BPs) such as pamidronate to increase OvCA cell elimination. (**B**) Strategies to enhance NK cell antitumor activities in OvCA. (i) Adoptive transferred NK cells, primed with IL-2, IL-15, and IL-18 to increase antitumor activities against OvCA via the production of granzymes, perforin, and IFNγ. (ii) Generation of IFNγ-, granzyme-, and perforin-producing NK cells by the administration of cytokine cocktails (IL-2, IL-15, and IL-18) to bone marrow or peripheral blood or cord blood-derived hematopoietic stem cells (HSCs); (iii) CD24/CD133 chimeric antigen receptor (CAR)-engineered NK cells with increased cytotoxic activity against CD24-positive OvCA cell lines (SKOV3 and OVCAR3).

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
