# Peer review of "The Ovarian Cancer Tumor Immune Microenvironment (TIME) as Target for Therapy: A Focus on Innate Immunity Cells as Therapeutic Effectors"

_ijms, 2020, doi:10.3390/ijms21093125_

Round 1

Reviewer 1 Report

This manuscript aims to systemically review the innate immunity related immune therapeutic approaches currently developed in the ovarian cancer. This review is comprehensive and should be interesting or informative for readers of this field. This review should be acceptable and most figures and tables were very helpful to realize the immunotherapy of ovarian cancer.

Author Response

We thank the reviewer for the positive comments.

Reviewer 2 Report

Comprehensive, well cited review of the field. Included figures are excellent and facilitate understanding.

-Accessible to specialist immune/ovarian cancer readership but may be less accessible for the broader readership of IJMS. Consider revising with this is mind and also inclusion of a glossary eg. ascites, serous, innate, adaptive, definition of different immune cells TANs, NETs, anergy, omental, decidua, DAMPs etc etc

Consider at the end of the conclusions section – inclusion of bullet point outstanding questions or future directions

Specific comments:-

Lose the hyphen between tumour and immune in the title – changes meaning.

-Include brief general introduction to the TME before focus on the role of immune cells in the TME which is one aspect of the ovarian cancer TME

-Quite a number of English language errors, requires careful proofreading by a native speaker of English to help communication.

Just a few examples below, but others:-

Line 33 – ‘or contrasting’ – unclear what is meant here

Line 47 – ‘associated to’ should be associate with

Line 75 – ‘apart for’ should be ‘apart from’

Line 104 – recurring more deepen investigation – unclear

Line 106  - change to ‘targets’

Line 108 monocytes

Line 120 cancer cells’ proliferation

Line 138 based to  - delete to

Line 139 through…

Line 157, line 165

Line 175 developed

Line 177 word missing seen to

Line 190 evidence now shows

Line 221 an

Line 224 word missing

Line252 shown rather than showed

Line 271 typo iii

Line 402 as to for?

Line 436 word order/choice

etc

Line 27 – ‘the crucial orchestrator’ – overstatement, many other contributory elements of the TME to ovarian cancer progression

Line 40/41 – state it is the third most common after… rather than one of the most common…

Line 41 – 2nd sentence –first part of sentence contradicts previous sentence; be more specific on overall prognosis and mortality

Line 77 ‘great attention devoted to the host as a target’ – oversimplification – revise sentence to reflect more the TME as a target rather than cancer cells

Lines 81-92 – there is reference to the peritoneal TME and a favourable metastatic soil and accumulation of maliganant ascites/massive production of ascites – the authors should make clear to the readership the source of ascites/causes and also (include a schematic overview) of how ovarian cancer and its TME changes as the cancer progresses and metastases (through stages referred to in lines 44-52)

line 118 – word missing? Increased M1/M2 ratio?

Figure 1 – typos in legend eg. VEFF, phagocitosis, multiple (ii) under B,C)

Figure 2- phagocytosis spelling

Author Response

Comprehensive, well cited review of the field. Included figures are excellent and facilitate understanding.

REPLY: We thank the reviewer for the positive comments.

Accessible to specialist immune/ovarian cancer readership but may be less accessible for the broader readership of IJMS. Consider revising with this is mind and also inclusion of a glossary eg. ascites, serous, innate, adaptive, definition of different immune cells TANs, NETs, anergy, omental, decidua, DAMPs etc etc.

REPLY: Many of the immunology concept are basic (anergy) and the acronym (such as TANs, NET) are spelled at first appearance and detailed in the list of Abbreviations at the end of artcile. Therefore, the journal does not have a section for glossary, but a list for abbreviations.

Consider at the end of the conclusions section – inclusion of bullet point outstanding questions or future directions.

REPLY: Conclusions in the revised version now stress more on the possible combination between immunotherapy and “classic” therapy against OvCA. Indeed, we have explained and detailed the critical points of OvCA therapy in summary, and in particular the tumor-induced immunosuppression and tolerance towards innate immune cells and the relevance to boost up the anti-tumor activity of the immune system.

Lose the hyphen between tumour and immune in the title – changes meaning.

REPLY: We agree with the reviewer's suggestion and therefore hyphen has been removed.

Include brief general introduction to the TME before focus on the role of immune cells in the TME which is one aspect of the ovarian cancer TME.

REPLY: The focus of our review is the TIME, to address originality to the manuscript and dealing with the immunotherapy sections. Therefore, we integrated a brief introductory part on TME, before going to the TIME.

Quite a number of English language errors, requires careful proofreading by a native speaker of English to help communication.

REPLY: We apologize for the various grammatical errors in English, and now the manuscript has been checked for typos and English quality.

Lines 81-92 – there is reference to the peritoneal TME and a favourable metastatic soil and accumulation of maliganant ascites/massive production of ascites – the authors should make clear to the readership the source of ascites/causes and also (include a schematic overview) of how ovarian cancer and its TME changes as the cancer progresses and metastases (through stages referred to in lines 44-52).

REPLY: We thank the reviewer for the comments. The section discussing the TME in OvCA in the new version has been improved.

Reviewer 3 Report

All comments and suggestions are in the attached PDF.

The authors should focus on 3 main problems:

  1. Editing for English
  2. Adding more detail. The review is broad, but the details are thin.
  3. Avoid informal, non-scientific, or misleading/inaccurate terms (especially "silent killer").

OVERALL REVIEW:
Baci et al present a review focused on roles of macrophages, neutrophils, γδ T-cells, and NK cells in the ovarian cancer tumor microenvironment. The manuscript is broad in scope, but not deep. It provides a basic introduction to the OvCa tumor-immune microenvironment (TIME) for those who do not know much about the topic. It is a paper that would help to introduce a reader like me, who is not an immunologist, to the topic. While I would probably want more details from this paper, I could use it as a reference, and then go to the citations to find more specific information.

STRENGTHS:
1. Provides broad, useful information about the tumor-immune microenvironment. I’d like the info here to be deeper and more detailed, but readers can easily look into the referenced papers to get deeper information.
2. Generally well organized, with the exception of a paragraph in the macrophage section. Despite the many corrections that are needed for English language and style, the paper is fairly easy to understand.
3. Nice figures, although they are a bit complex. Perhaps a legend to the side showing each cell type and a label would be helpful?

WEAKNESSES:
1. Needs major revision for English. I have listed many examples below of corrections that need to be made. However, there are many more. I suggest the authors hire an English-language scientific editor to fix the issues, or at the very least they should give it to native English speakers/writers to make corrections.
2. More detail needed in many places, especially when the authors discuss previous animal experiments. I’d like to see more detail given about exactly what cell lines were used, what subtype of ovarian cancer they represent, where xenograft models came from, and what mouse strains were used.
3. In some places, the authors use rather informal and vague language (e.g. Line 220 “waking up dormant cancer cells”). Maybe this is a result of English translation problems? No matter the cause, these vague or informal phrases should be replaced with more professional, scientific phrases and more detail so that the reader can better understand the meaning.
4. Some sections are out of place. For example, the paragraph at Lines 165-183 is about PD-L1 and related therapies, but it’s in a section about macrophages. The authors need to do a better job of showing why the PD-L1 paragraph belongs in the macrophage section
5. Figure 1 is missing some labels (described below).

SPECIFIC CRITIQUES / SUGGESTED CHANGES AND CORRECTIONS:
Line 25: “more and more” needs re-write

Line 26: “it is now clear that the immune contexture in OvCA, here referred to as Tumor Immune Microenvironment (TIME), acts as the crucial orchestrator of OvCa progression” This may be an overstatement, and should probably be softened. Many other factors contribute to progression (e.g. genetic background of the patient and the tumor, other TME cells such as adipocytes, etc.). I agree that the immune environment is a very important orchestrator, but if the authors want to suggest that it is the most important factor, then they need to back this statement up with specific data.
Line 27: OvCa or OvCA? Pick one.
Line 28: “nowadays” is highly informal. Use a different phrase.
Line 33: “contrasting?” What does this mean? Is this the right word?
Line 33: should “TME” be “TIME?”
Line 47: “associated to diagnosis” should be “associated with diagnosis”
Line 51: (THIS IS A MAJOR FLAW) “Not a coincidence, OvCA has been dubbed the silent killer.” This sentence is sensationalist, trite, and inaccurate. It has no place in a scientific paper. It only causes fear and does nothing to educate us. There are many known symptoms of ovarian cancer. We must do our best to educate the public, health care providers, and patients so that they know the symptoms and can seek the proper care as early as possible. The authors should remove this sentence. Or, better yet, they should replace it with something useful, such as listing the symptoms and clinical signs that ARE present in patients, and the steps that patients and gynecologists can take to make a proper diagnosis.
Line 61: The treatment described (debulking surgery followed by platinum/taxol chemo) is the standard of care for nearly all patients, not just “early diagnosed.”
Line 65: period between words needs to be removed “anti-angiogenic. agents”
Line 75: “Apart for” should be changed to “Apart from”
Lines 102-106: This entire paragraph needs a rewrite or editing for clarity. For example, on Line 104 the statement “those recurring more deepen investigation” doesn’t make sense.
Line 105: “gd T cells” should be changed to the appropriate Greek letters
Line 108: “monocyte are enrolled” should be “monocytes are recruited”?
Line 112: “signals, as IL-4” should be “signals, such as IL-4”
Line 124: The legend below Figure 1 describes labels such as (i) and (ii), but these labels are missing on the actual image.
Line 125-137: The legend for Figure 1 contains a lot of acronyms that need to be introduced in the text.
Line 133: “VEFF” should be “VEGF”
Line 155: “consists in the retrieval of” needs re-write. Should be “consists of the retrieval?”

Line 159: “the T cell” should be just “T cell”
Line 161: “percentage” should be “percent”
Line 165-183: This entire paragraph is nice info, but it’s not clear from the text how it relates to macrophages, which are the topic of this section (3. Macrophages). The authors need to do a better job of showing specific links between PD-L1 and other therapies to the topic of macrophages if they want to keep this paragraph in this section. As an alternative, the authors could create a separate section dedicated only to clinical trials and other potential therapies. Then the sections about specific cell types could focus only on basic science topics.
Line 165: there should be a space in “PD-L1immune”
Line 171: “Avelumab is” should be “Avelumab was”
Line 177: “been seen correlate with” should just be “correlates with”
Line 182: “also CD47 has” should be “CD47 has also”
Line 190: “Evidences now show” should be “Evidence now shows”?
Line 200: “Releasing” should be “release”
Line 205: There is mention here of the role of neutrophils in facilitating extravasation of tumor cells. The authors should note that ovarian cancer cells do not spread throughout the pelvic/peritoneal cavity by the same process as most other solid tumors. Those other tumors require extravasation and then spread through vasculature or lymphatics. Ovarian cancer cells do not have to do this because already have direct access to the entire abdominal cavity. This is somewhat covered by mentioning in the next paragraph (Lines 211-223) that neutrophils can help establish the pre-metastatic omental niche. However, it is an important feature/distinction that is somewhat unique to ovarian cancer that it can spread to such organs without going through the vasculature or lymphatics, and this distinction may not be obvious to all readers.
Line 220: What does “waking up dormant cancer cells” mean? As with some other phrases, this is too vague and informal. Please use more scientific language, be specific in meaning, and give more detail.
Line 221: “essential” should be “an essential”
Line 227: “correlate” should be “correlates”
Line 240: “effect, combing” should be “effect of combining” or “effect when combining”
Line 248: “on the role” should be “of the role”
Line 252: “showed” should be “shown” or “demonstrated”
Line 287: “the innate immunity” should be just “innate immunity”

Line 305: The acronym “PB” is introduced, but the authors do not first state what it means. Later, on Line 310, they show that it means “peripheral blood.” The authors need to introduce the term on Line 305 the first time they use it.
Line 307: “the pamidronate” should be just “pamidronate”
Line 309: “These are in lines” should be “These data are in line”
Line 311: Maybe it’s an error in the PDF, but there appears to be an extra space on this line between “higher” and “drug”
Line 318: “grazymes” should be “granzymes”
Line 327: What does “Apart for the necessary tolerogenic behavior” mean? I understand that there is immune tolerance in this situation, but the phrase is odd. Re-phrase for clarity.
Line 341: “Treg cells increases in OvCA TME and release large amount” should instead be “Treg cells are increased in the OvCA TME and release large amounts”?
Line 348: Ovarian cancer is a broad term for many different diseases. The authors should clarify what types of ovarian cancer these cell lines represent. OVCAR3 and CAOV-3 are probably high-grade serous, but SKOV3 and A2780 are certainly not (for reference see the Domcke et al. 2013 paper in Nature Communications; DOI: 10.1038/ncomms3126). I suggest the authors also be cautious and specific in other parts of the paper when referring to experiments that have been done, and what cell lines or other disease models were used. They should state clearly each time if the models were for high-grade serous, clear cell, or some other ovarian cancer type. It would be very helpful for the reader if the authors could summarize in a Table how each of the different immune cell types (macrophages, gd T-cells, NK cells, etc.) is functioning in the TME of each subtype. It would be an excellent quick reference for readers.
Line 349-351: Again, what subtype was this xenograft model? Where were the tumor cells given (IP, subcutaneous, flank, etc.)? What kind of mice were used? There are many places in this paper where the information is too thin. MORE DETAIL!!!
Line 352: “larger” should be “higher”
Line 356: remove the words “long-lastly”
Line 357: “dampen” should be “dampens”
Line 358: “Large amount of MUC16 has” should be replaced with “Large amounts of MUC16 have” or “High levels of MUC16 have”
Line 361: “heathy” should be “healthy”
Line 364: “contribute” should be “contributes”
Line 371: “compare” should be “compared”

Line 374: “have” should be “has”
Line 385: several extra spaces between “to” and “increase”
Line 386: Sentence starting with “Concern in the use of IL-1…” does not make sense. Needs re-write.
Line 393: Re-write first sentence of this paragraph for clarity. English editor needed.
Line 398: “elevate” should be “elevated”
Line 402: “As for to IL-2” is not needed at the beginning of the sentence.
Line 402-409: Entire paragraph needs editing for English and clarity.
Line 412: “persist on” should be “persist in”
Line 412: “iL-15” should be capitalized
Line 414: “amount” should be “amounts”
Line 418: “has been also” should be “have also been”
Line 422: More detail. What’s the model? Where did the tumor cells come from? What mouse strain?
Line 432: The authors use the abbreviation “OC” to denote ovarian cancer. But they haven’t used OC anywhere else in the paper. Instead, they have used “OvCA.” Please use only one abbreviation and be consistent.
Line 435: First sentence of this paragraph needs re-write for clarity.

Author Response

OVERALL REVIEW:

Baci et al present a review focused on roles of macrophages, neutrophils, γδ T-cells, and NK cells in the ovarian cancer tumor microenvironment. The manuscript is broad in scope, but not deep. It provides a basic introduction to the OvCa tumor-immune microenvironment (TIME) for those who do not know much about the topic. It is a paper that would help to introduce a reader like me, who is not an immunologist, to the topic. While I would probably want more details from this paper, I could use it as a reference, and then go to the citations to find more specific information.

STRENGTHS:

  1. Provides broad, useful information about the tumor-immune microenvironment. I’d like the info here to be deeper and more detailed, but readers can easily look into the referenced papers to get deeper information.
  2. Generally well organized, with the exception of a paragraph in the macrophage section. Despite the many corrections that are needed for English language and style, the paper is fairly easy to understand.
  3. Nice figures, although they are a bit complex. Perhaps a legend to the side showing each cell type and a label would be helpful?

REPLY: We thank the reviewer for the positive comments and immediately after our reply point-by-point.

WEAKNESSES:

Needs major revision for English. I have listed many examples below of corrections that need to be made

REPLY: Manuscript has been checked for typos and English quality.

Generally, well organized, with the exception of a paragraph in the macrophage section.

REPLY: The Macrophages section has been completely revised and improved.

Nice figures, although they are a bit complex. Perhaps a legend to the side showing each cell type and a label would be helpful?

REPLY: Cell type is indicated close the specific icon, for each figure. A legend to the side showing each cell type and a label would generate confusion, being the figures already reach of necessary details.

More detail needed in many places, especially when the authors discuss previous animal experiments. I’d like to see more detail given about exactly what cell lines were used, what subtype of ovarian cancer they represent, where xenograft models came from, and what mouse strains were used.

REPLY: We agree with the reviewer’s comment. Even if for every statement the proper references are provided, we indicate, what cell lines were used, what subtype of ovarian cancer they represent, at least at their first appearance.

In some places, the authors use rather informal and vague language (e.g. Line 220 “waking up dormant cancer cells”).

REPLY: Now in the revised manuscript all informal and vague terms have been replaced with more technical terms.

Some sections are out of place. For example, the paragraph at Lines 165-183 is about PD-L1 and related therapies, but it’s in a section about macrophages. The authors need to do a better job of showing why the PD-L1 paragraph belongs in the macrophage section.

REPLY: Thank you for this criticism, and now the Macrophages section has been completely revised and improved.

Line 124: The legend below Figure 1 describes labels such as (i) and (ii), but these labels are missing on the actual image.

REPLY: The matches between (i), (ii), (iii) between Figure 1 and related legend have been integrated in the revised version.

SPECIFIC CRITIQUES / SUGGESTED CHANGES AND CORRECTIONS:

Line 25: “more and more” needs re-write…………..

REPLY: We thank the reviewer for all the criticisms and suggestions that we have followed in order to eliminate all inaccuracies in the revised ms.

Line 165-183: This entire paragraph is nice info, but it’s not clear from the text how it relates to macrophages, which are the topic of this section (3. Macrophages). The authors need to do a better job of showing specific links between PD-L1 and other therapies to the topic of macrophages if they want to keep this paragraph in this section. As an alternative, the authors could create a separate section dedicated only to clinical trials and other potential therapies. Then the sections about specific cell types could focus only on basic science topics.

REPLY: The Macrophages section has been completely revised and improved.

Line 205: There is mention here of the role of neutrophils in facilitating extravasation of tumor cells. The authors should note that ovarian cancer cells do not spread throughout the pelvic/peritoneal cavity by the same process as most other solid tumors. Those other tumors require extravasation and then spread through vasculature or lymphatics. Ovarian cancer cells do not have to do this because already have direct access to the entire abdominal cavity. This is somewhat covered by mentioning in the next paragraph (Lines 211-223) that neutrophils can help establish the pre-metastatic omental niche. However, it is an important feature/distinction that is somewhat unique to ovarian cancer that it can spread to such organs without going through the vasculature or lymphatics, and this distinction may not be obvious to all readers.

REPLY: We thank the reviewer for the comments and knowledge of relevant issue on OvCA spread. This part has been mentioned, as a peculiar feature of OvCA.

Round 2

Reviewer 3 Report

The authors did an excellent job of revising the original text to add more detail and to remove informal language. This is now a strong review that is written in a professional style. The text still needs extensive editing for English language. The suggested changes are included in the attached PDF. I don't like being the unpaid English editor for this review, but it seems that no one else will do it. If the authors make the suggested changes for syntax and clarity, then the paper will be ready to publish. I am satisfied with the scientific content of the paper. It's a good review. Please make the changes for language and go ahead with publishing. I do not need to see another revision.

Author Response

We thank the reviewer for positive comments, following first revision.

We also are very excited for the positive comments on the revised section for Macrophages.

We address all the new comments, that allow us to further improve the quality of the review.

Comments  on the scientific sections, proposed corrections and the suggested edit for English improvements, have been addressed.

Finally, we would like to thank the reviewer  for the strong efforts in editing our text.